# Top observables as precise probes of the ALP

**Anh-Vu Phan**[1,2⋆]

**1** Institute for Mathematics, Astrophysics and Particle Physics, Radboud University, 6500 GL Nijmegen, The Netherlands
**2** Nikhef, Science Park 105, 1098 XG Amsterdam, The Netherlands

⋆ anhvu.phan@ru.nl

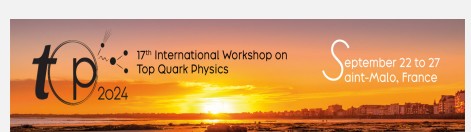

## Abstract

**Measurements of the top quark by the ATLAS and CMS experiments go beyond testing the Standard Model (SM) with high precision. Axion-like particles (ALPs), a potential SM extension involving new pseudoscalar particles, exhibit strong interactions with heavy SM fermions. Consequently, they can significantly affect the kinematic distributions of top quarks in top-antitop pair production. Moreover, such strong interactions can induce other ALP couplings at low energies, leading to a rich phenomenology. We summarize recent developments in probing the ALP-top coupling and use LHC data from run 2 to constrain the ALP parameter space.**

## 1  Axion-like particles

The axion-like particle (ALP), denoted as $a$, is a pseudo Nambu-Goldstone boson whose interactions with the Standard Model (SM) particles preserve the shift symmetry $a \to a + 2\pi f_a$, where $f_a$ is related to the cutoff scale $\Lambda$ via $\Lambda = 4\pi f_a$. This light degree of freedom may arise from new physics at energy above the cutoff scale. While inspired by the QCD axion, originally proposed to resolve the strong CP problem [1,2], the ALP framework is more general. Unlike the QCD axion, the ALP models do not aim to address the strong CP problem and thus imposes no relation between the ALP mass and its couplings. The Lagrangian of the ALP effective theory is [3]

$$\mathcal{L}_{\text{eff}}^{I}(\mu) \supset \frac{1}{2}\partial_\mu a \partial^\mu a - \frac{m_a^2}{2}a^2 - \frac{\partial^\mu a}{f_a}\sum_F \bar{F}\mathbf{c}_F\gamma_\mu F + \frac{a}{f_a}\sum_V c_{VV}\frac{\alpha_V}{4\pi}V_{\mu\nu}^A\tilde{V}^{A,\mu\nu}. \tag{1}$$

Here, we adopt the notation used in [4]. $F$ and $V$ represent the SM fermions and vector bosons, respectively. The last two terms, which describe the interaction between the ALP and the SM particles, are invariant under the shift symmetry. However, we allow for the possibility

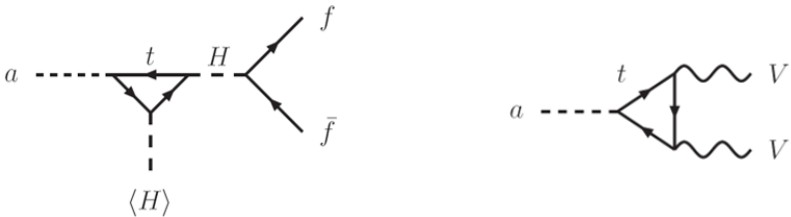

Figure 1: Representative diagrams that source the running of $c_{ff}$ (left) and $\tilde{c}_{VV}$ (right). Figures taken from [4].

of breaking this symmetry by including the mass term for the ALP. We refer to this formulation as *basis I*, as indicated by the corresponding superscript.

In our analysis, it is often more convenient to use an alternative form of the above Lagrangian, derived through an appropriate chiral transformation [5], which we refer to as *basis II*,

$$\mathcal{L}_{\text{eff}}^{II}(\mu) \supset \frac{1}{2}\partial_\mu a \partial^\mu a - \frac{m_a^2}{2}a^2 - \frac{a}{f_a}\left(\bar{Q}\tilde{H}\tilde{\mathbf{Y}}_U U + \text{h.c.}\right) + \tilde{c}_{GG}\frac{\alpha_s}{4\pi}\frac{a}{f_a}G_{\mu\nu}^a\tilde{G}^{\mu\nu,a}. \tag{2}$$

Here, only the terms relevant for top observables at the LHC are shown. $\tilde{H} = i\sigma_2 H$ with $H$ being the SM Higgs doublet, and

$$\tilde{\mathbf{Y}}_U = i\left(\mathbf{Y}_U\mathbf{c}_U - \mathbf{c}_Q\mathbf{Y}_U\right), \tag{3}$$

$$\tilde{c}_{GG} = c_{GG} + \frac{1}{2}\text{Tr}\left(\mathbf{c}_U + \mathbf{c}_D - 2\mathbf{c}_Q\right). \tag{4}$$

The couplings $\mathbf{c}_Q$, $\mathbf{c}_U$, $\mathbf{c}_D$, and $c_{GG}$ are the ones that appear in (1). This basis explicitly highlights that the ALP-quark interaction is proportional to the quark Yukawa coupling $\mathbf{Y}_U$, a feature not readily apparent from (1). This makes the top quark an especially promising candidate for probing the ALP. To focus exclusively on the top-quark sector, we neglect ALP couplings to other SM particles, retaining only those to the top quark and gluons. Under these assumptions, and in the basis where the quark Yukawa matrix is diagonal, we have

$$\left(\tilde{\mathbf{Y}}_U\right)_{33} = iy_t\left(\mathbf{c}_U - \mathbf{c}_Q\right)_{33} \equiv iy_t c_{tt}, \tag{5}$$

$$\tilde{c}_{GG} = c_{GG} + \frac{c_{tt}}{2}, \tag{6}$$

while the other elements of the $\tilde{\mathbf{Y}}_U$ matrix vanish.

## 2   ALP phenomenology

Due to the renormalization group (RG) running of the couplings sourced by loop diagrams such as those shown in Fig. 1, the ALP-top coupling $c_{tt}$ can induce other couplings at lower energy scales, resulting in a rich phenomenology. Specifically, the ALP couplings at a scale $\mu$ are related to those at the cutoff scale $\Lambda > \mu$ by [5]

$$c_{tt}(\mu) = \left[1 - \frac{9}{2}R(\mu, \Lambda)\right]c_{tt}(\Lambda), \tag{7}$$

$$\tilde{c}_{GG}(\mu) = \tilde{c}_{GG}(\Lambda) - R(\mu, \Lambda)c_{tt}, \tag{8}$$

with the evolution function

$$R(\mu, \Lambda) = \frac{\alpha_t(\mu)}{\alpha_s(\mu)}\left[1 - \left(\frac{\alpha_s(\Lambda)}{\alpha_s(\mu)}\right)^{\frac{1}{7}}\right], \qquad \alpha_t(\mu) = \frac{y_t^2(\mu)}{4\pi}. \tag{9}$$

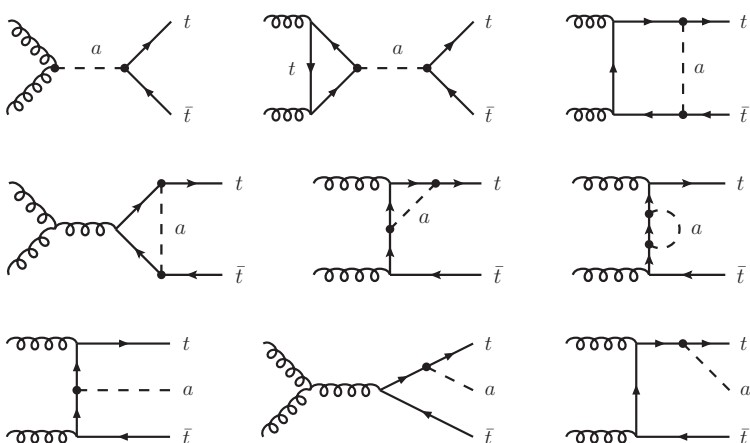

Figure 2: Representative diagrams for ALP contributions to $t\bar{t}$ production. The first row shows UV-finite diagrams. Counterterms are required to renormalize UV-divergent diagram in the second row. Diagrams with real ALP emission are shown in the last row. Figures taken from [4].

For example, if $c_{tt}(\Lambda) = 1$ and $\tilde{c}_{GG}(\Lambda) = 0$ at $\Lambda = 4\pi$ TeV, the values of the couplings at the scale $\mu = m_t = 172.5$ GeV are [4]

$$c_{tt}(m_t) = 0.81, \quad \tilde{c}_{GG}(m_t) = -0.04. \tag{10}$$

Unless otherwise specified, throughout the remainder of this article, we assume all couplings between the ALP and SM particles are set to zero at the cutoff scale, except for the ALP-top coupling. Due to RG running, these couplings become nonzero at lower energy scales, giving rise to a wide variety of observable signatures.

The lifetime of the ALP determines the nature of its signatures in experimental searches. Through top-quark loops, an ALP can decay into $\gamma\gamma$, $e^+e^-$, $\mu^+\mu^-$, or hadrons, depending on its mass [6]. ALPs with masses below approximately 100 MeV are best searched for in invisible decays of mesons [7]. For heavier ALPs with masses up to $\sim 10$ GeV, displaced vertex searches are more sensitive [6,7]. At the LHC, events involving a pair of top quarks produced in association with a displaced ALP can serve as triggers for these searches [6]. For ALPs with masses above this range but below the $t\bar{t}$ production threshold, indirect probes through virtual corrections to SM processes provide the most widely used strategy [4,8–11]. Finally, for ALP masses exceeding $2m_t$, resonance searches become feasible [12].

## 3   ALPs in $t\bar{t}$ production

To probe ALPs with mass between 10 GeV and $2m_t$, we analyze their corrections to $t\bar{t}$ observables. At tree-level, ALPs can modify the $gg \to t\bar{t}$ process via an s-channel diagram, such as the first diagram shown in Fig. 2. This contribution, however, requires both $c_{tt}$ and $\tilde{c}_{GG}$ to be nonzero. For $\tilde{c}_{GG}(\Lambda) = 0$, the tree-level diagram still contributes at scales $\mu < \Lambda$ through an induced gluon coupling, as computed via (8). In this scenario, its contribution is of the same order as the other ALP-corrected loop diagrams depicted in the first two rows of Fig. 2. To account for potential high-energy enhancements, we also include the squared contribution of the first diagram.

The diagrams in the second row are UV-divergent and require renormalization, as detailed in [4]. Additionally, since real ALP emissions are unresolved in the inclusive top observables considered here, the diagrams in the third row of Fig. 2 must also be included.

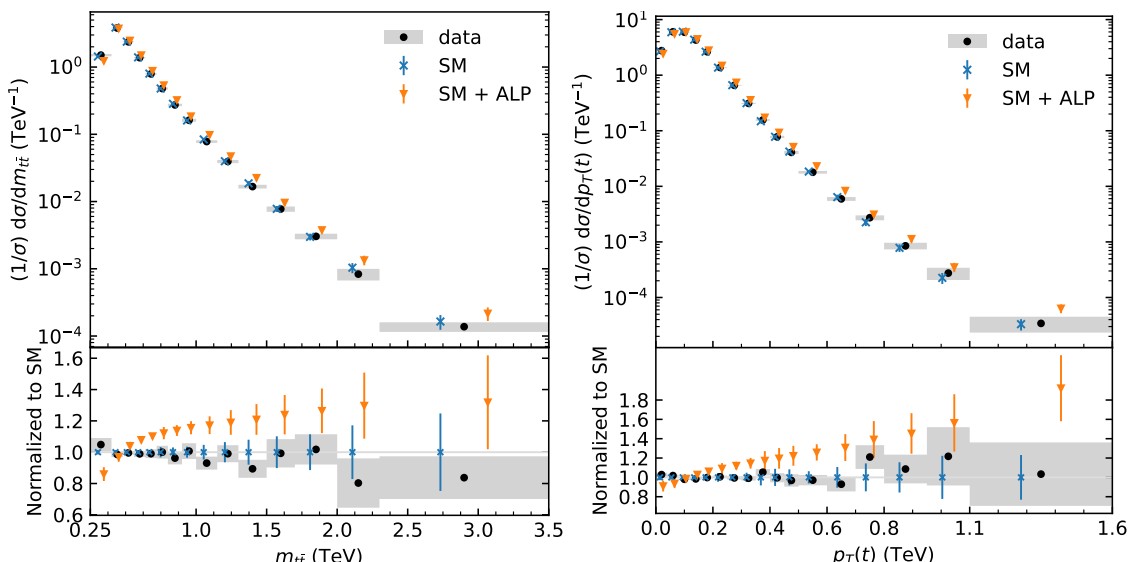

Figure 3: Normalized $t\bar{t}$ invariant mass $m_{t\bar{t}}$ distribution (left) and normalized transverse momentum $p_T(t)$ distribution of the hadronically decaying top (right) for $t\bar{t}$ production in the SM (blue) and in the SM with ALPs (orange) for $c_{tt}(\Lambda)/f_a = 20/$TeV, $c_{GG}(\Lambda) = 0$, and $m_a = 10$ GeV. Theoretical uncertainties are represented by vertical bars. Experimental data from CMS [13] are shown as black dots with uncertainty bands in grey. Figures taken from [4].

## 4  Comparison with experimental results

To probe ALPs with mass between 10 GeV and $2m_t$, we examine their corrections to $t\bar{t}$ observables. In our analysis, the renormalization and factorization scales are set at a dynamical scale

$$\mu_F = \mu_R = \mu = \frac{1}{2}\sum_{f=t,\bar{t}} m_T(f), \qquad m_T^2(f) = m_f^2 + p_T^2(f), \tag{11}$$

where $f = t, \bar{t}$ are top and antitop quarks at parton level and $m_T(f)$ is their transverse mass. As shown in Fig. 3, the presence of an ALP modifies the kinematic distributions of top-antitop production. A detailed analysis can be found in [4]. Among the distributions studied, $m_{t\bar{t}}$ and $p_T(t)$ distributions are most sensitive to ALP corrections. In both distributions, the overall ALP contribution is negative, suppressing the normalized distributions at low energy, while at high energy, ALPs enhance the distributions. In the $m_{t\bar{t}}$ distribution, this enhancement arises primarily from real ALP emission, while in the $p_T(t)$ distribution, the momentum dependence of ALP-gluon coupling enhances the s-channel tree-level contribution to the cross section at high energy. For a detailed discussion of ALP effects on angular distributions, see [4].

To obtain a bound on the ALP couplings $c_{tt}$ and $c_{GG}$, we fit our theoretical predictions with data from CMS [13], taking into account both theoretical and experimental uncertainties. We find that for $c_{GG}(\Lambda) = 0$, the invariant mass distribution is better suited to probe the ALP-top coupling. Although both the $m_{t\bar{t}}$ and $p_T(t)$ distributions are enhanced at high energy, the large theoretical and statistical uncertainties in the tails of these distributions mean that these energy enhancements do not significantly impact the fitting results. This, however, may not hold in the future, as more precise measurements of the high-energy tails become available from ATLAS and CMS.

Our bounds at 95% C.L. are shown in Fig. 4. In the left panel, we set $c_{GG}(\Lambda) = 0$ and obtain the bounds $|c_{tt}(\Lambda)/f_a| < 11.3/$TeV with $\Lambda = 4\pi$ TeV for $m_a \lesssim 200$ GeV for a combined

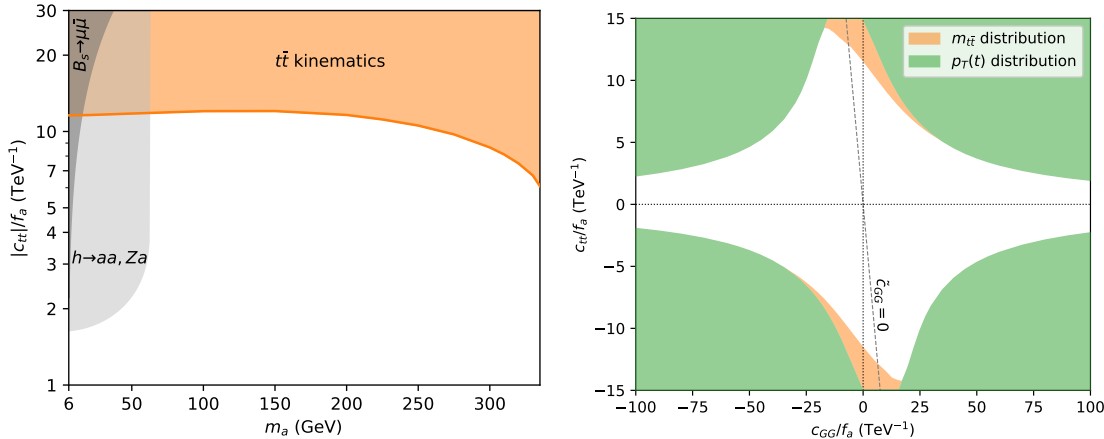

Figure 4: Left: constraints on $c_{tt}(\Lambda)/f_a$ for $c_{GG}(\Lambda) = 0$ at 95% C.L. from our fit to the CMS $t\bar{t}$ production kinematic distribution [13]. Constraints from $B_s \to \mu\bar{\mu}$ [7,14] and Higgs decay $h \to aa, Za$ [15,16] are shown in dark grey and light grey, respectively. Right: Bounds on $c_{tt}(\Lambda)/f_a$ and $c_{GG}(\Lambda)/f_a$ for $0 < m_a \lesssim 200$ GeV obtained by fitting the $m_{t\bar{t}}$ (orange) or $p_T(t)$ (green) distributions with [13]. Figures taken from [4].

fit of all bins in the $m_{t\bar{t}}$ distribution. We do not include in this fit other kinematic distributions to avoid double-counting. This bound is strengthen for ALP masses close to the $t\bar{t}$ production threshold, where ALP resonance production can occur. To study the effect of the ALP-gluon coupling, we keep $m_a \lesssim 200$ GeV and fit both $c_{tt}(\Lambda)$ and $c_{GG}(\Lambda)$. The constraints are weakest near $\tilde{c}_{GG}(\Lambda) = 0$ due to a cancellation of contributions from the first and the second diagrams in Fig. 2 in basis I. For small values of $c_{GG}(\Lambda)$, the invariant mass distribution is most sensitive. For large $c_{GG}(\Lambda)$, the s-channel tree level squared contribution grows quadratically with $c_{GG}$, dominating over real ALP emission, which makes the $p_T(t)$ distribution more sensitive. Larger values of $c_{GG}(\Lambda)$ are more effectively probed by dijet production [10].

## 5 Conclusion and outlook

ALPs offer a rich phenomenology due to their shift-symmetric couplings and sensitivity to the top quark. This study analyzes ALP effects on $t\bar{t}$ production kinematics, incorporating tree-level and loop-induced contributions, and constrains ALP-top and ALP-gluon couplings using CMS data. For $c_{GG}(\Lambda) = 0$, we obtained $|c_{tt}(\Lambda)/f_a| < 11.3/\text{TeV}$ for $m_a \lesssim 200$ GeV. These bounds are consistent with those from other analyses [9, 17, 18], while covering a broader mass range and including the effects of the ALP-gluon coupling. This approach is phenomenologically relevant for $10$ GeV $\lesssim m_a < 2m_t$. Lighter ALPs are better probed in invisible decay modes of mesons [7] and displaced ALP decay vertices [6], while ALPs with $m_a > 2m_t$ are more suited to resonance searches in top-antitop production [12].

Given the nature of ALPs, top spin observables present promising topics for future analyses. Upcoming high-luminosity LHC runs will improve the precision to top observables, allowing for increased sensitivity to virtual effects of ALPs. A future top factory, such as the FCC-ee, will simultaneously serve as an ALP factory, providing unprecedented statistics and precision. Furthermore, the renormalization group running of ALP couplings suggests that improved measurements of meson decays or better knowledge of the Higgs decay width and its exotic decay channels all contribute to probing light new physics, such as ALPs, through virtual effects.

## Acknowledgements

We would like to thank the organizers of the Top 2024 workshop for organizing the workshop and for providing us with the opportunity to present our work.

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
