# Peer review of "Top observables as precise probes of the ALP"

_SciPost Physics Proceedings_

## Round 1 · Referee Report · Anonymous (Referee 1) · 2025-1-9

Strengths

1-Well written, clear summary of work presented at the conference
2-The original calculation is an important contribution to the study of ALP phenomenology at colliders
3-Quality of the work is clear as the original article has already been published in a high-impact peer-reviewed journal

Weaknesses

No major weaknesses.

Report

The paper is concise & well written, summarising the original work well and what was presented at the conference (https://indico.cern.ch/event/1368706/contributions/6012425/attachments/2933131/5151719/Top-of-the-ALPs.pdf) . In my opinion, the article meets all of the criteria for a conference proceedings and I recommend its publication. I note a few minor points that could improve the discussion slightly below.

Requested changes

1- page 2, in the discussion after eqn (4), it is remarked that in this basis, the ALP-top coupling is proportional to the top Quark yukawa coupling, justifying the special phenomenological interest in top-ALP couplings by the fact that ithis quantity is large. This is a bit confusing because both bases must be equivalent as the second is obtained from the first by an ALP-dependent chiral transformation. They must therefore give identical physical predictions. Perhaps the discussion could be modified to emphasise that the second basis highlights the fact that top quark ALP interactions may be of special interest. At the moment it reads like the second basis is somehow different.

2- eqn (7), the quantity R(mu, Lambda) is not defined

3- in the presentatioin of the result, the renormalisation and factorisation scale choices should be explicitly stated.

Recommendation

Publish (easily meets expectations and criteria for this Journal; among top 50%)

---

## Round 2 · Author Response

My apologies for the wrong arxiv number.

---

## Round 2 · List of Changes

The correct arxiv number is used

---

## Editorial Decision

accepted_in_target_journal